# Molecular and microbiological evidence of bacterial contamination of intraocular lenses commonly used in canine cataract surgery

**Kourtney K. Dowler**[1], **Aida Vientós-Plotts**[1], **Elizabeth A. Giuliano**[1], **Zachary L. McAdams**[2], **Carol R. Reinero**[1], **Aaron C. Ericsson**[3]*

**1** Department of Small Animal Medicine and Surgery, College of Veterinary Medicine, University of Missouri, Columbia, Missouri, United States of America, **2** Molecular Pathogenesis and Therapeutics Program, University of Missouri, Columbia, Missouri, United States of America, **3** Department of Veterinary Pathobiology, College of Veterinary Medicine, University of Missouri, Columbia, Missouri, United States of America

* ericssona@missouri.edu

**Data Availability Statement:** All 16S rRNA amplicon sequencing data have been deposited in the National Center for Biotechnology Information

## Abstract

Inflammatory outcomes, including toxic anterior segment syndrome (TASS) and infectious endophthalmitis, are potentially painful, blinding complications following cataract surgery. In an in vitro pilot study, commercially available, sterile foldable intraocular lenses (IOLs) used during routine canine cataract surgery, and their packaging fluid were surveyed for the presence of bacterial DNA and/or viable (cultivable) bacteria. Swabs from IOLs and packaging fluid from three different veterinary manufacturers and three different production lots/manufacturer were collected for 16S ribosomal ribonucleic acid (rRNA) sequencing. Packaging fluid samples were collected for aerobic/capnophilic bacterial culture. Culture yielded one isolate, identified as *Staphylococcus epidermidis*. 16S rRNA sequencing revealed distinct brand-specific bacterial DNA profiles, conserved between IOLs and packaging fluid of all production lots within each manufacturer. The dominant taxonomy differentiating each manufacturer was annotated as *Staphylococcus* sp, and was a 100% match to *S. epidermidis*. Distinct mixtures of bacterial DNA are present and consistent in IOLs and packaging fluid depending on the manufacturer, and *Staphylococcus* is the dominant contributor to the bacterial DNA detected. Caralens products had a significantly lower amount of *Staphylococcus spp*. compared to Anvision and Dioptrix products.

## Introduction

Cataracts are the leading cause of blindness in both dogs and people and are a common reason dogs are referred to veterinary ophthalmologists worldwide [1, 2]. Phacoemulsification remains the gold standard for cataract removal in both human and veterinary ophthalmology and is performed daily by specialists in the USA to restore patient vision [3, 4]. Intraocular lens implantation is essential to provide an optimal visual outcome in both people and dogs.

Undesirable inflammatory outcomes, including toxic anterior segment syndrome (TASS) and infectious endophthalmitis, are potentially painful, blinding complications following

(NCBI) Sequence Read Archive (SRA) under BioProject ID PRJNA729058. All code and sample metadata can be accessed on GitHub (https://github.com/ericsson-lab/IOL).

**Funding:** KD - ACVO Vision for Animals Foundation (VAF grant 2021-3)(www.visionforanimals.org) and the MU Phi Zeta chapter. The funders had no role in study design, data collection and analysis, decision to publish, or preparation of the manuscript.

**Competing interests:** The authors have declared no competing interests exist.

cataract surgery. TASS is a sterile postoperative inflammatory reaction believed to be caused by a noninfectious substance that enters the anterior segment, resulting in toxic damage to the intraocular tissues [5]. This phenomenon is noted in physician ophthalmology and often spurs a thorough investigation of sterilization procedures, operating room equipment, and analysis of fluids and medications used during surgery [5–11]. A primary differential diagnosis for TASS is infectious endophthalmitis. In a report by Ledbetter *et. al.*, infectious endophthalmitis developed in 0.15% of canine eyes undergoing phacoemulsification over a 20-year study period. Bacterial culture of aqueous and vitreous humor samples recovered *Staphylococcus* and *Streptococcus* species [12]. Importantly, our current understanding of TASS and infectious endophthalmitis have been significantly limited to culture-dependent techniques.

A recently published report described the occurrence of a specific form of fibrinous uveitis (fibrin web, FW) in a population of dogs that underwent unilateral or bilateral phacoemulsification at the University of Missouri Veterinary Health Center (MU-VHC) between 2014–2018 [13]. This FW was observed emanating from the intraocular lens (IOL) and capsular bag into the anterior chamber. It bears some similarity to anterior chamber fibrinoid syndrome after cataract extraction, as described in people [14].

In the original manuscript in dogs [13], pre- and peri-operative factors associated with FW formation were evaluated and the authors described the long-term effects on canine patient vision and comfort. Based on the available data, IOL implantation, viscoelastic type, patient age, and total phacoemulsification time were associated with FW formation. Diabetes mellitus, sex, cataract stage, surgeon, intracameral injections other than viscoelastic, and intra- and postoperative complications were not associated with FW formation. The findings of this publication raised obvious, and as yet unanswered, questions as to the underlying cause of this specific phenomenon with direct translational implications [13].

Recently, molecular alternatives to identify microbial communities based on conserved regions in bacterial genomes have been developed [15–17]. These alternatives allow for culture-independent microbial analysis, even in low biomass samples [18]. Currently, there are limited reports in both physician and veterinary literature regarding microbial analysis of commonly used, sterile commercial products designated for use during cataract surgery and their relationship to endophthalmitis (specifically *Pseudomonas* spp.) or TASS outbreaks following surgery [19–21]. The available reports utilized conventional bacterial culture techniques for microbial analysis. Numerous physician reports have identified the presence of a biofilm (specifically *Staphylococcus* spp.) on the surface of intraocular lenses as a potential cause for endophthalmitis following cataract surgery with IOL implantation [22–31]. To the authors' knowledge, there are no reports investigating this phenomenon in veterinary ophthalmology.

This study aimed to investigate common, commercially available, sterile foldable IOLs and their packaging fluid used during routine canine cataract surgery for the presence of bacterial DNA and/or viable (cultivable) bacteria. We hypothesized that (1) a microbiota exists for sterile foldable IOLs and their packaging fluid, (2) the microbial communities will differ among different IOL manufacturers, (3) *Pseudomonas* and/or *Staphylococcus*, will be identified as the dominant genus, and (4) this microbial population will not be detected with standard culture techniques.

## Materials and methods

Sterile, foldable IOLs and their sterile packaging fluid were obtained from three different veterinary manufacturers (Anvision, Dioptrix, and Caralens) and three different production lots/manufacturer were collected for a total of 27 IOL samples and 27 packaging fluid samples.

## Conventional bacterial culture

Packaging fluid samples were collected for conventional culture concurrently with microbiota samples. For Anvision and Dioptrix products, packaging was opened carefully on a sterile field so as to avoid contamination of the packaging fluid. Due to the package design of CaraLens products, the IOL was first removed from the packaging and placed in a sterile 5 mL tube prior to aspiration of packaging fluid. Approximately 0.5 mL of fluid was aspirated aseptically using a sterile 22 ga needle and syringe and then dispensed onto a separate culture swab (ESwab[TM], COPAN Diagnostics Inc, Murrieta, CA). A separate swab was utilized for each culture. Samples for culture were temporarily stored at 4° Celsius before transmission and submission to the University of Missouri Veterinary Medical Diagnostic Laboratory for aerobic and capnophilic bacterial culture. Samples were plated directly onto blood and MacConkey agar for aerobic growth at 35°C in ambient air. An additional blood agar plate was utilized and stored at 35°C in a capnophilic jar. Bacterial identification was performed as previously described [32].

## DNA extraction

For Anvision and Dioptrix products, packaging was opened carefully on a sterile field so as to avoid contamination of the packaging fluid, which was aspirated aseptically using a sterile 22 ga needle and syringe. 750 µL of fluid was dispensed into a bead tube from a QIAamp Power-Fecal kit (Qiagen, Hilden, Germany) for DNA extraction. The IOLs were then carefully removed from the packaging using sterile surgical instruments, placed in a sterile 5 mL polypropylene tube containing 800 µL lysis buffer [33], sealed, and placed on a rocker at room temperature for 15 minutes. Up to 750 µL of lysis buffer was then transferred to a bead tube from a QIAamp PowerFecal kit. Due to the package design of CaraLens products, the IOL was first removed from the packaging and placed in a sterile 5 mL tube prior to aspiration of packaging fluid. Negative (i.e., unused culture swabs and lysis buffer) controls were included in the DNA extraction process, alongside the study samples, and analyzed. The negative controls included unused e-swab tips and lysis buffer that did not make contact with the IOLs or irrigating fluid.

Bead tubes were then agitated for three minutes at 30 Hz using a TissueLyser II (Qiagen), placed on a 70°C heat block for 15 minutes, and then handled according to the manufacturer's instructions. Eluted DNA was quantified using Qubit 2.0 fluorometer (Invitrogen) and Quant-iT HS dsDNA assay kits. As all samples yielded scant DNA, samples were concentrated using a Sorvall SpeedVac for 30–60 minutes to achieve the minimal volume needed for 16S rRNA library preparation.

## 16S rRNA amplicon library generation and sequencing

The V4 hypervariable region of the 16S rRNA gene was amplified from extracted DNA using dual-indexed universal primers (U515F/806R) [34] with flanking Illumina adapters and the TruSeq UDI adapter design. The polymerase chain reaction used for amplification was as follows: $98°C^{(3\ min)} + [98°C^{(15\ sec)} + 50°C^{(30\ sec)} + 72°C^{(30\ sec)}] \times 25$ cycles $+ 72°C^{(7\ min)}$ [35]. The 16S rRNA V4 amplicon libraries were pooled and sequenced using an Illumina MiSeq System with V2 chemistry generating 2×250 paired end reads.

## Informatics

Informatics were performed with Quantitative Insights Into Microbial Ecology 2 (QIIME2) v2021.2 [36]. Briefly, sequences were trimmed of the Illumina adapters with cutadapt [37]. Using DADA2 [38], trimmed forward and reverse reads were truncated to 150 base pairs,

paired, then denoised into unique sequences called Amplicon Sequence Variants (ASVs). Due to the low biomass of these samples, features identified as contaminants using the decontam [39] prevalence method with default settings were filtered from the dataset, thus, the retained ASVs were considered to be truly present in the intraocular lenses and packaging fluid. A feature table containing the frequency of each ASV per sample was rarefied to 6,067 total features per sample maximizing the number of subsampled features per sample and total number of samples retained for further analysis. Samples with a total feature number less than 6,067 were omitted from downstream analyses. Taxonomy was assigned to each unique ASV with a sklearn algorithim [40] using the QIIME2-provided 99% non-redundant SIVLA v138 reference database of the 515F/806R region of the 16S rRNA gene [36, 41]. Code used for sequence processing can be accessed at https://github.com/ericsson-lab/IOL.

## Statistical analysis

Coverage was compared between IOL and fluid of each of the manufacturers using a Mann-Whitney rank sum test. Two-way permutational analysis of variance (PERMANOVA) test using Bray-Curtis and Jaccard similarities, fourth-root transformation, and principal coordinate analyses (PCoAs) of the rarefied feature table were performed using the vegan [42] package v2.5.7 within R [43] v4.1.2. The complete annotated feature table with assigned taxonomy was used to construct an ASV-level relative taxonomic abundance plot. One- and two-way analysis of variance (ANOVA) and heatmap generation using Euclidian distance measures with a Ward clustering algorithm of ASV relative abundance were performed using the open-access software MetaboAnalyst 5.0 [44].

## Results

### Conventional bacterial culture

One of 27 packaging fluid samples yielded *Staphylococcus epidermidis* from an Anvision sample via conventional bacterial aerobic and capnophilic culture. No growth of this organism was obtained on direct culture, rather it was recovered from enrichment broth. Of note, this sample had the highest read count for *Staphylococcus* spp. of all IOL and packaging fluid samples in the DNA sequencing analysis. No other packaging fluid samples yielded aerobic or capnophilic growth. Collectively, we believe that these data indicate producer-specific microbial DNA residues in commercially available veterinary IOLs, dominated by Staphylococcal organisms, and with the potential for contamination with viable bacterial cells.

### DNA sequencing

There were no significant differences in coverage when comparing the IOL and packaging fluid from the same manufacturer (Table 1). Swabs and lysis buffer samples yielded varying number of reads ranging from 35492 to 236663 (median 175274). Subjective assessment of the

**Table 1. Read counts for each intraocular lens/packaging fluid manufacturer.**

|  | IOL (median, range) | Fluid (median, range) | *p-value* |
|---|---|---|---|
| AnVision | 15728 (603–203658) | 42513 (6940–211357) | 0.077 |
| CaraLens | 35086 (16554–256665) | 16909 (6813–77213) | 0.077 |
| Dioptrix | 33705 (4443–83114) | 16520 (3623–159169) | 0.791 |

There were no significant differences in coverage when comparing the intraocular lens and fluid from the same manufacturer. ($p < 0.05$)

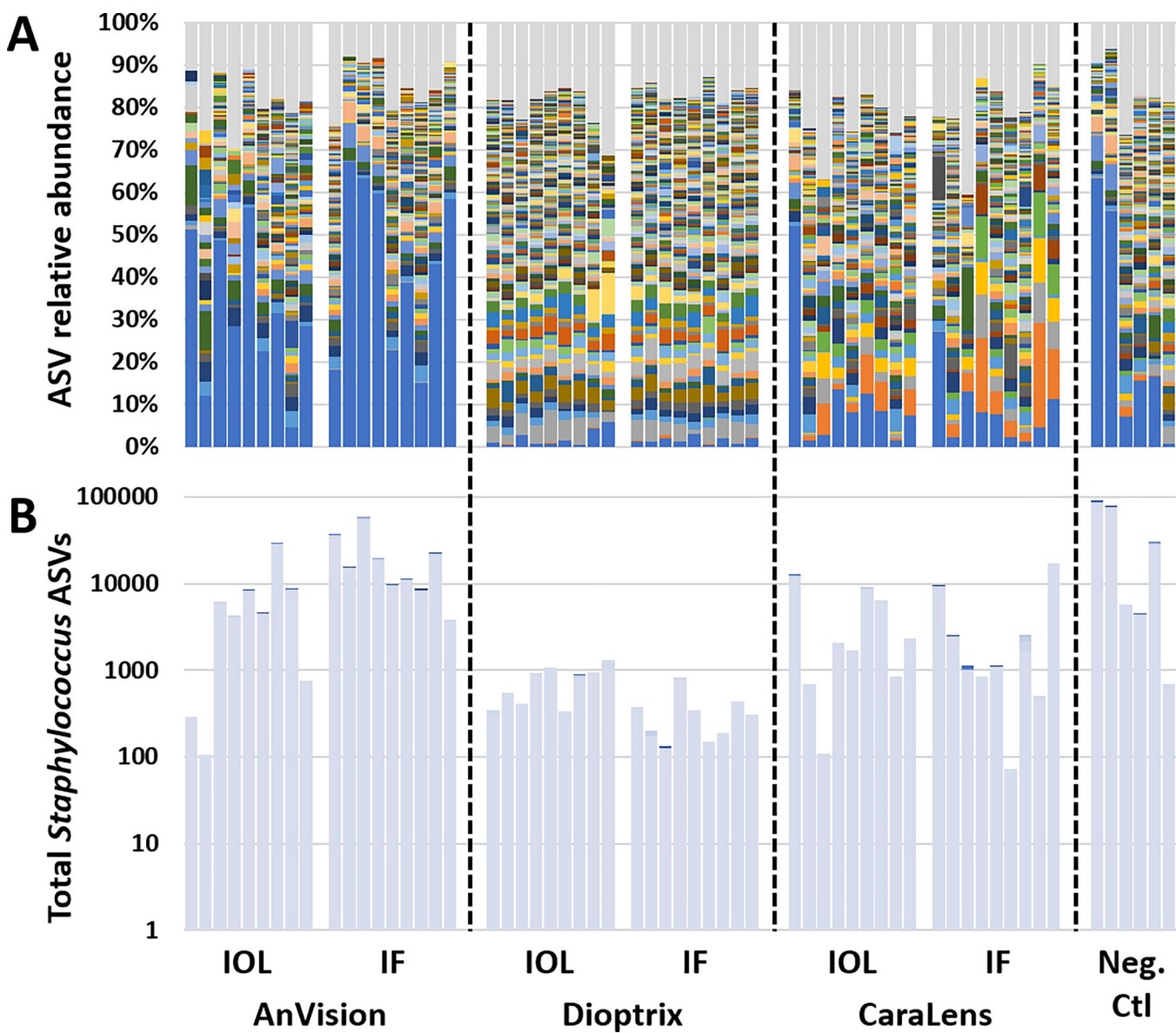

**Fig 1. Composition of each manufacturer.** Intraocular lenses (IOL) and packaging fluid yield varying microbial taxonomic profiles (A). Relative taxonomic abundance plots of the genus *Staphylococcus* for IOL/packaging fluid samples (according to manufacturer) and negative controls. (B).

community structure of each manufacturer revealed distinct mixtures of bacterial DNA present in the IOLs and packaging fluid (Fig 1A). The IOL and packaging fluid were consistent within each manufacturer. *Staphylococcus* spp. was the dominant contributor to the bacterial DNA detected (particularly in Anvision and, to a lesser degree, Dioptrix samples). (Fig 1B) Bacterial genera representing greater than a 5% mean relative abundance were Staphylococcus in AnVision and Dioptrix IOL and PF samples and Bacillus in Dioptrix PF samples (S1 Table).

Principal coordinate analyses (PCoA) yielded brand-specific microbial communities independent of sample type (Fig 2). Pairwise comparisons of samples by brand using Bray Curtis (BC) and Jaccard (J) similarities indicated differences between Anvision and Caralens (BC: $p \leq 0.001$, J: $p \leq 0.001$), Anvision and Dioptrix (BC: $p \leq 0.001$, J: $p \leq 0.001$), and Caralens and Dioptrix (BC: $p \leq 0.001$, J: $p \leq 0.001$) (Tables 2 and 3). Pairwise comparisons of samples by

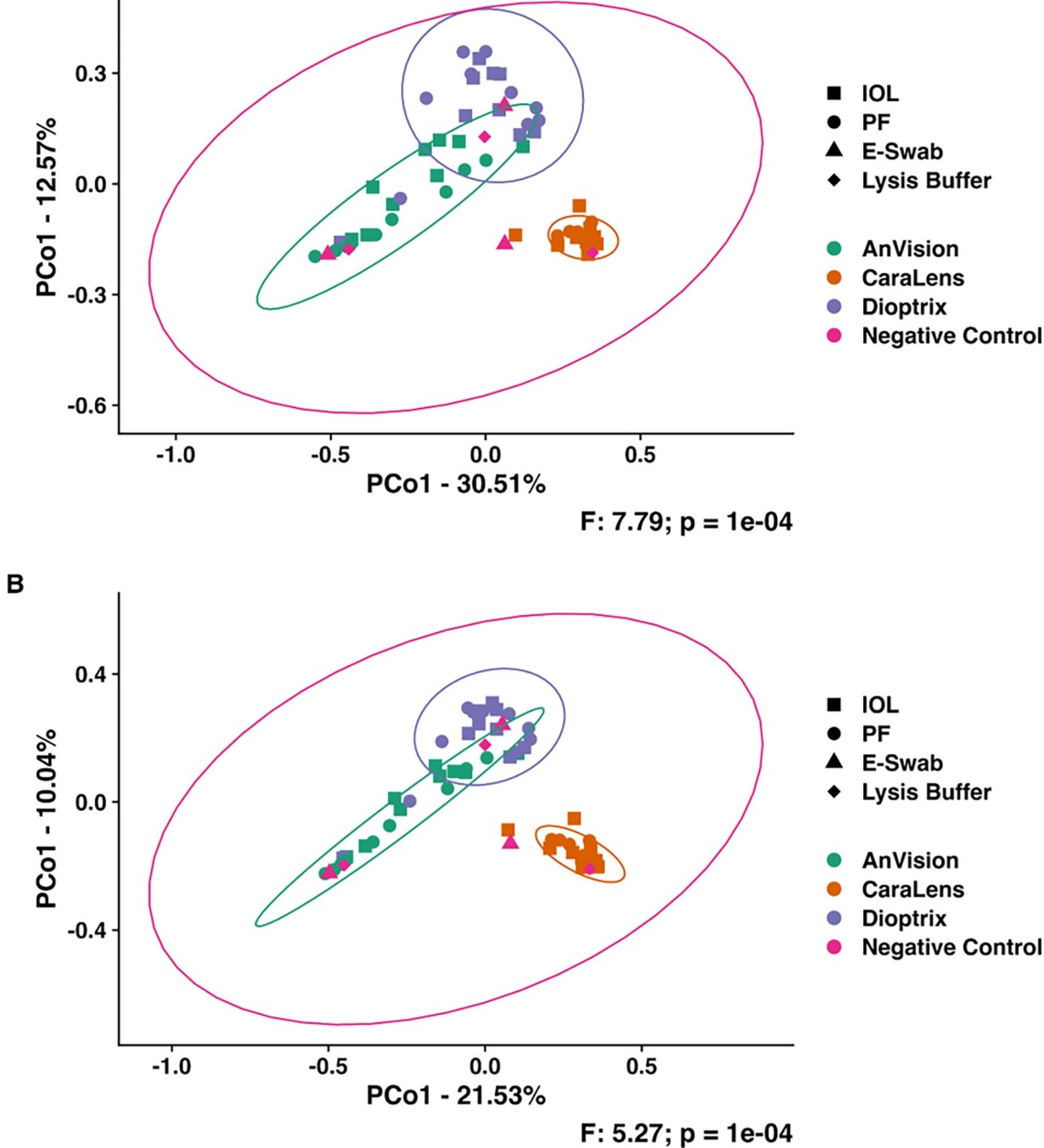

**Fig 2. Principal coordinate analyses of each IOL manufacturer.** Brand-specific clustering of microbial communities for intraocular lens (■), packaging fluid (●), E-swabs (▲), and lysis buffer (◆) were found according to manufacturer. PCoAs using (A) Bray Curtis.and (B) Jaccard distances with 95% confidence ellipses.

sample source using Bray Curtis (BC) and Jaccard (J) similarities indicated no differences between the IOL and packaging fluid groups (BC: *p = 0.7277*, J: *p = 0.7384*).

A two-way ANOVA test of the feature table found no differentially abundant ASVs due to sample type (packaging fluid or IOL) but 110 ASVs, yielding an FDR-corrected

**Table 2. Pairwise comparisons by brand using Bray Curtis similarities.**

|  | AnVision | CaraLens | Dioptrix | Negative Control |
|---|---|---|---|---|
| AnVision |  | **0.0002** | **0.0002** | 0.267 |
| CaraLens | 19.40 |  | **0.0002** | **0.0005** |
| Dioptrix | 5.926 | 11.66 |  | 0.0463 |
| Negative Control | 1.040 | 5.395 | 1.811 |  |

P-values (upper right) and F values (lower left) generated from one—way PERMANOVA of Bray Curtis similarity indices between microbial DNA detected in each of the brands and negative controls. ($p < 0.05$)

p-value $< 0.05$, out of 3,538 total ASVs differing due to sample brand with no interactions. A heatmap of the 110 significantly differing ASVs revealed brand-specific clusters of increased ASV abundance with no clustering by sample type (Fig 3). Two ASVs were found at an average abundance greater than 5%. ASV0001 (*Staphylococcus* soo.) was significantly abundant in IOL and PF samples from AnVision (31.66% ± 18.57%, 42.69% ± 20.49%, respectively), and Dioptrix (12.94% ± 16.20% and 9.41% ± 8.08%, respectively). ASV0002 (*Bacillus* spp.) was also significantly abundant in Dioptrix PF (8.52% ± 8.64%).

## Discussion

The present study demonstrated that 16S rRNA sequencing revealed distinct brand-specific bacterial DNA profiles, conserved between IOL and packaging fluid of all production lots within each manufacturer. Culture yielded one isolate, identified as *Staphylococcus epidermidis*. The dominant taxonomy differentiating each manufacturer was annotated as *Staphylococcus* sp. and was a 100% match to *S. epidermidis*.

The negative controls (e.g., unused culture swabs, lysis buffer solution) utilized in this study closely resembled the IOL and packaging fluid samples, and no analyzed sample was truly devoid of bacterial DNA. Despite this, the distinct clustering of IOLs and packaging fluid (with multiple production lots), as shown in the PCoAs and stacked bar chart, along with separation of samples from different manufacturers is compelling. The disparities between manufacturers identified in our data are likely attributable to the inherent differences in those products from different IOL manufacturers. The dominant amplicon sequence variants (ASV) included microbes from the genera *Staphylococcus* in AnVision and Dioptrix samples and *Bacillus* in Dioptrix samples. Distinct clustering of the most abundant ASVs found in the 3 different brands indicated dissimilar community compositions (Fig 3), and subjectively, Caralens had a greater mixed population of organisms while Anvision and Dioptrix had lesser variation in their populations. The apparent differences feasibly originate in the particular production process of each IOL brand, as the microbiota clustering is conserved between the IOL and its respective packaging fluid.

**Table 3. Pairwise comparisons by brand using Jaccard similarities.**

|  | AnVision | CaraLens | Dioptrix | Negative Control |
|---|---|---|---|---|
| AnVision |  | **0.0002** | **0.0002** | 0.267 |
| CaraLens | 11.88 |  | **0.0002** | **0.0011** |
| Dioptrix | 4.123 | 7.744 |  | 0.0428 |
| Negative Control | 1.040 | 3.542 | 1.526 |  |

P-values (upper right) and F values (lower left) generated from one—way PERMANOVA of Jaccard similarity indices between microbial DNA detected in each of the brands and negative controls. ($p < 0.05$)

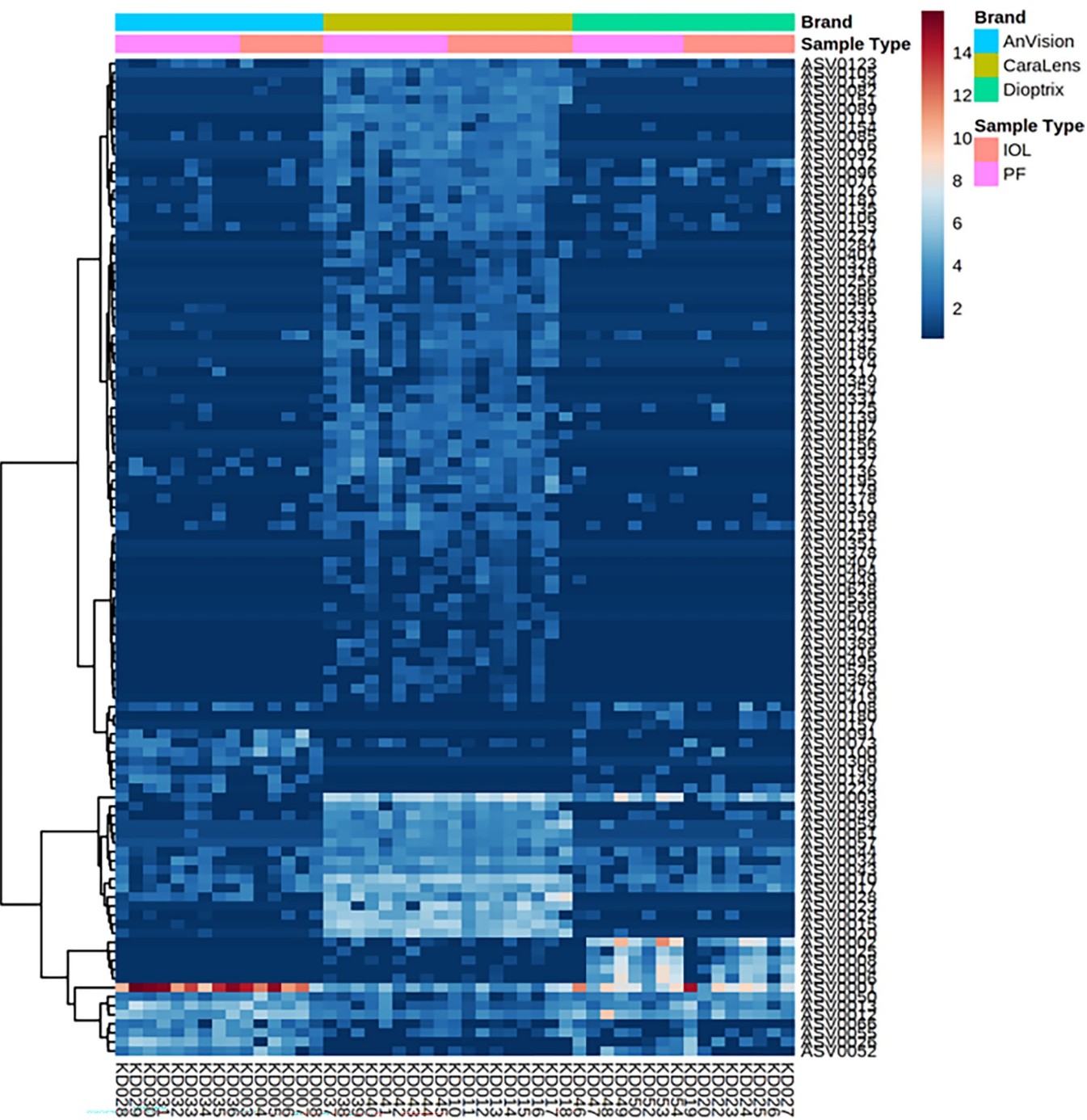

**Fig 3. A heatmap of the 110 significantly differing amplicon sequence variants.** Distinct clustering of the most abundant sequence variants was found in the 3 different brands, indicating dissimilar community compositions.

The *Staphylococcus* nucleotide sequence in our study was compared to the database manually. *S. epidermidis* was 100% match (along with *S. warneri*, *S. aureus*, and few other less common species). This *Staphylococcus* species may have originated from processes during our DNA extraction or inoculation of culture media. However, careful measures were taken to avoid human contamination during the extraction process, including utilizing sterile

technique (i.e., sterile gown, gloves, and draping) to place the IOLs and packaging fluid in Eppendorf tubes and obtain the conventional bacterial cultures. The subsequent tubes used for extraction were handled with gloved hands taking care not to contaminate the samples and internal walls of the tubes during the pipetting processes. It is unlikely that the lysis buffer was the source of Staphylococcal contamination as our samples demonstrated dissimilar contamination amongst the manufacturers (Fig 1). The Staphylococcal contamination in the Anvision and Dioptrix products was not significantly different from the negative controls, but that of Caralens was significantly different from the two other manufacturers and the negative controls (Tables 2 and 3). This supports the idea that the lysis buffer used for DNA extraction in all samples was not the source of the identified Staphylococcal contamination. Another plausible explanation is that it was a skin commensal contaminant on the production line where IOLs are manufactured and packaged. Although we cannot fully rule out the possibility of Staphylococcal contamination during handling of the samples, we believe our findings highlight the importance to not dismiss the presence of Staphylococcal species in these investigations.

Limited reports regarding microbial analysis of commercial products for use during physician cataract surgery and their relationship to post-operative infectious endophthalmitis outbreaks have utilized conventional culture techniques to identify the source of contamination. An acute post-cataract surgery endophthalmitis outbreak investigation in India revealed that aqueous and vitreous humor samples cultured *Pseudomonas aeruginosa*, along with the hydrophilic acrylic intraocular lenses and their solution [19]. The isolates from the patients and the intraocular lens packaging solution revealed matching patterns similar to a strain of *P. aeruginosa* on polymerase chain reaction (PCR) analysis [19]. An investigation of an outbreak of multidrug-resistant Pseudomonas aeruginosa endophthalmitis following cataract surgery revealed contaminated trypan blue solutions used during surgery [20].

One limitation of our study was that it was in vitro investigation with a relatively small sample size. The number of IOLs per manufacturer was chosen in an attempt to reflect the true community composition of the individual manufacturer, but a power analysis was not performed. DNA sequencing cannot distinguish between viable and non-viable bacteria, but bacterial DNA may still be present in the samples. This bacterial DNA can be inflammatory on its own, or other bacterial products may be present (e.g., lipopolysaccharide, peptidoglycan, or other various lipopeptides). The lack of viability does not rule out the potential to induce pathologic changes via immunologic mechanisms. Toll-like receptor 9-mediated mechanisms involve recognition of nucleic acids derived from bacteria and viruses, leading to production of type 1 interferons and pro-inflammatory cytokines [45]. Theoretically, immune recognition of this introduced bacterial DNA and/or bacterial products can result in intraocular inflammation following cataract surgery. Various TASS outbreak investigations have identified viscoelastics, trypan blue, inadequate sterilization procedures, and intraocular lenses as the source of TASS [46–51]. One physician report described a TASS outbreak that was strongly associated with bacterial biofilm contamination of autoclave reservoirs, identified by culture. The authors proposed the transport of heat-stable bacterial cell antigens in steam onto the surgical instrumentation as the inciting cause of the TASS-like clinical signs in the patients [48]. TASS investigations have also identified endotoxin as an inciting cause for the outbreak [11, 48, 52, 53]. Endotoxins (e.g., heat-stable lipopolysaccharide) may not be deactivated by the autoclave sterilization process, and even small amounts can cause TASS [10]. If endotoxin enters the eye, it may cause severe anterior segment inflammation [53]. The United States Pharmacopeia recommends two testing methods for endotoxin: the gel-clot method and the kinetic method [5]. These methods were not utilized in this study, but we believe it is possible that bacterial products, such as lipopolysaccharide, could be present in our samples. Historical TASS outbreaks

utilized conventional bacterial culture to rule out infectious endophthalmitis, so it is possible that culture-independent microbial analyses of future outbreaks may yield clinically relevant bacterial DNA profiles.

There were no significant differences in the IOL and packaging fluid bacterial read counts according to each of the three manufacturers. There was also no significant difference in the bacterial community composition of the IOL and packaging fluid within each manufacturer, indicating a brand-specific microbial population. Future studies could either investigate the IOL or packaging fluid to globally reflect the brand-specific bacterial community structure. Additional in vitro microbiota investigations may include analysis of various other intracameral agents used during routine cataract surgery, such as viscoelastic products, trypan blue, or even preservative-free dexamethasone. Future in vivo studies may involve a microbiota investigation of aqueous humor samples in post-phacoemulsification patients with and without FW development. The data in our study lead us to consider the possibility that the presence of a microbiota population inherent in each individual IOL manufacturer may be an inciting cause for FW formation. A previous study found that canine eyes that received Anvision IOL/viscoelastic combination during cataract surgery had a higher prevalence of FW formation in the early post-operative period compared to other commercial IOL/viscoelastic combinations [13]. That finding combined with the data found in our study highlights the necessity for further investigation of FW formation prospectively in clinical patients.

This study provided important data regarding the presence of microbial populations in three brands of commercially available veterinary IOLs and their packaging fluid. We anticipate future applications of these data in studies regarding prediction or prevention of the development of a specific form of fibrinous uveitis following cataract surgery in dogs and people. Both TASS and endophthalmitis are devastating complications after routine cataract surgery that result in loss of vision and in many cases, necessitate enucleation. These data are vital to furthering our understanding of both post-operative endophthalmitis and TASS after cataract surgery and improving surgical outcomes. Our results have critically important translational relevance to both veterinary and physician ophthalmologists.

## Conclusions

Distinct mixtures of bacterial DNA are present and consistent in IOLs and packaging fluid depending on the manufacturer. *Staphylococcus* is the dominant contributor to the bacterial DNA detected in this in vitro pilot study, and the amount of Staphylococcal DNA found in Caralens products was significantly less compared to Anvision and Dioptrix.

## Supporting information

**S1 Table. Taxa present at ≥ 5% average abundance in intraocular lens and packaging fluid samples.** Average percentages and standard deviation of the most abundant bacterial groups for each manufacturer based on 16S rRNA sequencing.
(DOCX)

## Acknowledgments

The authors wish to thank Becky Dorfmeyer and Giedre Turner for their technical support during the course of this project.

## Author Contributions

**Conceptualization:** Kourtney K. Dowler, Aida Vientós-Plotts, Elizabeth A. Giuliano, Zachary L. McAdams, Carol R. Reinero, Aaron C. Ericsson.

**Data curation:** Kourtney K. Dowler, Zachary L. McAdams.

**Formal analysis:** Aida Vientós-Plotts, Zachary L. McAdams, Aaron C. Ericsson.

**Funding acquisition:** Kourtney K. Dowler, Elizabeth A. Giuliano.

**Investigation:** Kourtney K. Dowler, Aida Vientós-Plotts.

**Methodology:** Aida Vientós-Plotts, Zachary L. McAdams, Carol R. Reinero, Aaron C. Ericsson.

**Resources:** Elizabeth A. Giuliano.

**Supervision:** Aida Vientós-Plotts, Elizabeth A. Giuliano, Carol R. Reinero, Aaron C. Ericsson.

**Visualization:** Aida Vientós-Plotts, Zachary L. McAdams, Aaron C. Ericsson.

**Writing – original draft:** Kourtney K. Dowler, Aida Vientós-Plotts.

**Writing – review & editing:** Kourtney K. Dowler, Aida Vientós-Plotts, Elizabeth A. Giuliano, Zachary L. McAdams, Carol R. Reinero, Aaron C. Ericsson.

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
