## [Decision Letter · Decision Letter 0]

25 Mar 2022

PONE-D-21-37300Molecular and microbiological evidence of bacterial contamination of intraocular lenses commonly used in canine cataract surgeryPLOS ONE

Dear Dr. Ericsson,

Thank you for submitting your manuscript to PLOS ONE. After careful consideration, we feel that it has merit but does not fully meet PLOS ONE’s publication criteria as it currently stands. Therefore, we invite you to submit a revised version of the manuscript that addresses the points raised during the review process.

Please, answer to all the comments of the reviewer. Consider carefully the comments related to the statistical treatment of the results.

We look forward to receiving your revised manuscript.

Kind regards,

Paula V Morais, Ph.D

Academic Editor

PLOS ONE

Journal Requirements:

Reviewers' comments:

Reviewer's Responses to Questions

**Comments to the Author**

1. Is the manuscript technically sound, and do the data support the conclusions?

Reviewer #1: Partly

2. Has the statistical analysis been performed appropriately and rigorously? 

Reviewer #1: Yes

3. Have the authors made all data underlying the findings in their manuscript fully available?

Reviewer #1: Yes

4. Is the manuscript presented in an intelligible fashion and written in standard English?

Reviewer #1: Yes

5. Review Comments to the Author

Reviewer #1: This study investigated the presence of bacteria and/or bacterial DNA contamination in presumably sterile and commercially available canine IOLs and their packaging fluid. I appreciate your efforts in this important and emerging area of research. The manuscript is well-written, concise, and informative. The number of products sampled for each manufacturer is relatively small, but as a descriptive pilot study this is adequate. Further clarification regarding negative controls and data filtering methods are necessary before publication.

Materials & Methods:

For consistency, please have the order of experiments in the Materials & Methods match the Results. For example, if conventional bacterial culture is mentioned first in the Materials & Methods, please list it first in the Results.

Although the authors have used negative controls to assess the presence of environmental contaminants, which can significantly influence samples from sites with low microbial biomass, it is not clear if contaminant filtering analysis was conducted to discount the possibility of reagent/environmental contaminants impacting the results. Information regarding subtractive filtering should be provided in the manuscript. More information is provided below.

Results:

Table S1 – “Uncultured” bacterium is likely inappropriate terminology for microbiome data. Consider renaming to unidentified or unlisted bacterium.

Discussion:

243-249: As mentioned above, subtractive filtering to remove contaminants is important for low biomass samples, especially since the negative controls closely resemble the IOL and fluid packaging samples in this study. Several methods have been developed to identify and remove contaminating DNA, and this should be utilized and described in the manuscript. For example, Decontam is an open-source R package that identifies contaminants in microbiome data to censor contaminant taxa from the dataset allowing researchers to generate more accurate profiles of microbial communities. Following the use of any subtractive filtering technique, the read count of each sample should be assessed to determine whether the minimum cutoff based on the rarefaction curve has been met. Samples with very low read counts should be excluded from further analysis.

Davis NM, Proctor DM, Holmes SP, Relman DA, Callahan BJ. Simple statistical identification and removal of contaminant sequences in marker-gene and metagenomics data. Microbiome. 2018;6(1):226.

https://doi.org/10.1186/s40168-018-0605-2

Zinter MS, Mayday MY, Ryckman KK, Jelliffe-Pawlowski LL, DeRisi JL. Towards precision quantification of contamination in metagenomic sequencing experiments. Microbiome. 2019;7(1):62.

https://doi.org/10.1186/s40168-019-0678-6

6. PLOS authors have the option to publish the peer review history of their article (what does this mean?). If published, this will include your full peer review and any attached files.

Reviewer #1: No

---

## [Author Response · Author response to Decision Letter 0]

20 Apr 2022

The authors wish to sincerely thank the reviewer for their time and editorial suggestions to our manuscript. We sincerely appreciate the fact that the reviewer appears to be as enthusiastic about this data as we are. Our responses to each comment can be found below in blue. We have made all suggested changes in the manuscript where appropriate.

Review Comments to the Author

Reviewer #1: This study investigated the presence of bacteria and/or bacterial DNA contamination in presumably sterile and commercially available canine IOLs and their packaging fluid. I appreciate your efforts in this important and emerging area of research. The manuscript is well-written, concise, and informative. The number of products sampled for each manufacturer is relatively small, but as a descriptive pilot study this is adequate. Further clarification regarding negative controls and data filtering methods are necessary before publication.

Thank you kindly for your time and effort in reviewing our manuscript. Please see our comments below regarding the use of data filtering methods in our updated manuscript. 

Materials & Methods:

For consistency, please have the order of experiments in the Materials & Methods match the Results. For example, if conventional bacterial culture is mentioned first in the Materials & Methods, please list it first in the Results.

Thank you for bringing this to our attention, and we have modified the Results section to improve consistency of the manuscript.

Although the authors have used negative controls to assess the presence of environmental contaminants, which can significantly influence samples from sites with low microbial biomass, it is not clear if contaminant filtering analysis was conducted to discount the possibility of reagent/environmental contaminants impacting the results. Information regarding subtractive filtering should be provided in the manuscript. More information is provided below.

We concur, and thank you for pointing this out. As part of the revision process, we have utilized the decontam package to censor contaminant taxa from our dataset. The retained amplicon sequence variants are considered to be truly present in the intraocular lenses and packaging fluid. We have revised the data within the manuscript, figures, and tables to reflect the updated dataset. 

Results:

Table S1 – “Uncultured” bacterium is likely inappropriate terminology for microbiome data. Consider renaming to unidentified or unlisted bacterium.

Thank you for bringing this to our attention, and we have modified Table S1 accordingly. 

Discussion:

243-249: As mentioned above, subtractive filtering to remove contaminants is important for low biomass samples, especially since the negative controls closely resemble the IOL and fluid packaging samples in this study. Several methods have been developed to identify and remove contaminating DNA, and this should be utilized and described in the manuscript. For example, Decontam is an open-source R package that identifies contaminants in microbiome data to censor contaminant taxa from the dataset allowing researchers to generate more accurate profiles of microbial communities. Following the use of any subtractive filtering technique, the read count of each sample should be assessed to determine whether the minimum cutoff based on the rarefaction curve has been met. Samples with very low read counts should be excluded from further analysis.

Thank you for bringing this to our attention. Please see our comments above regarding the use of decontam to remove contaminant DNA in our dataset. Our data, manuscript, figures, and tables have been updated accordingly. Additionally, the reference (listed below) has been added to revised manuscript regarding the script used to filter out contaminant sequences. 

Davis NM, Proctor DM, Holmes SP, Relman DA, Callahan BJ. Simple statistical identification and removal of contaminant sequences in marker-gene and metagenomics data. Microbiome. 2018;6(1):226.

https://doi.org/10.1186/s40168-018-0605-2

---

## [Decision Letter · Decision Letter 1]

3 Nov 2022

Molecular and microbiological evidence of bacterial contamination of intraocular lenses commonly used in canine cataract surgery

PONE-D-21-37300R1

Dear Dr. Ericsson,

We’re pleased to inform you that your manuscript has been judged scientifically suitable for publication and will be formally accepted for publication once it meets all outstanding technical requirements.

Kind regards,

Paula V Morais, Ph.D

Academic Editor

PLOS ONE

Additional Editor Comments (optional):

Reviewers' comments:

Reviewer's Responses to Questions

**Comments to the Author**

1. If the authors have adequately addressed your comments raised in a previous round of review and you feel that this manuscript is now acceptable for publication, you may indicate that here to bypass the “Comments to the Author” section, enter your conflict of interest statement in the “Confidential to Editor” section, and submit your "Accept" recommendation.

Reviewer #1: All comments have been addressed

2. Is the manuscript technically sound, and do the data support the conclusions?

Reviewer #1: Yes

3. Has the statistical analysis been performed appropriately and rigorously? 

Reviewer #1: Yes

4. Have the authors made all data underlying the findings in their manuscript fully available?

Reviewer #1: Yes

5. Is the manuscript presented in an intelligible fashion and written in standard English?

Reviewer #1: Yes

6. Review Comments to the Author

Reviewer #1: Dear Authors,

Thank you for addressing my comments. I have no further concerns or comments prior to publication.

7. PLOS authors have the option to publish the peer review history of their article (what does this mean?). If published, this will include your full peer review and any attached files.

Reviewer #1: No

---

## [Editor Report · Acceptance letter]

8 Nov 2022

PONE-D-21-37300R1 

Molecular and microbiological evidence of bacterial contamination of intraocular lenses commonly used in canine cataract surgery 

Dear Dr. Ericsson:

I'm pleased to inform you that your manuscript has been deemed suitable for publication in PLOS ONE. Congratulations! Your manuscript is now with our production department. 

Kind regards, 

on behalf of

Professor Paula V Morais 

Academic Editor

PLOS ONE